# The Impact of Grouping on Skin Lesions and Meat Quality of Pig Carcasses

**DOI:** 10.3390/ani10040544

**Published:** 2020-03-25

**Authors:** Bert Driessen, Sanne Van Beirendonck, Johan Buyse

**Affiliations:** 1Research Group Animal Welfare, 3583 Paal, Belgium; bert.driessen@dierenwelzijn.eu; 2Laboratory of Livestock Physiology, Department of Biosystems, KU Leuven, 3001 Heverlee, Belgium; 3Bioengineering Technology TC, KU Leuven, 2440 Geel, Belgium; sanne.vanbeirendonck@kuleuven.be

**Keywords:** grouping, meat quality, pigs, skin lesions, transport

## Abstract

**Simple Summary:**

In practice, unfamiliar pigs are frequently mixed prior to loading in order to obtain groups of uniform weight and to adjust the group size to the dimensions of the trailer compartments. Regrouping pigs is associated with establishing a new social rank via aggressive interactions. Fighting results in skin lesions and pre-slaughter stress, which leads to reduced meat quality. In this study, four grouping strategies, namely, non-regrouping and regrouping at fattening (regrouped at 80 kg and kept till slaughter), loading and lairage, were compared by determining skin lesions and meat quality at slaughter. The non-regrouped pigs showed, at slaughter, fewer skin lesions and better meat quality than the pigs regrouped at loading or in lairage. Pigs mixed at 80 kg at the farm have, in general, a comparable amount of skin lesions and comparable meat quality as the non-mixed group. If mixing is unavoidable, due to large within-group weight variations, mixing at 80 kg can be an alternative to reduce skin lesions at slaughter and to optimise meat quality. However, mixing at 80 kg is still associated with aggressive interactions after regrouping and with weight variations at slaughter.

**Abstract:**

In practice, unfamiliar pigs are frequently mixed prior to loading in order to obtain groups of uniform weight and to adjust the group size to the dimensions of the trailer compartments. Mixing pigs induces aggressive interactions to establish a new social rank. Fighting results in skin lesions and pre-slaughter stress and, in turn, reduced meat quality. A study was performed to compare the effect of non-regrouping and regrouping at fattening (at 80 kg and kept till slaughter), loading and lairage. A total of 1332 pigs were included over 30 transports from one pig farm to one slaughterhouse (110 km). Skin lesions were determined on 1314 carcasses. Meat quality was measured on 620 pigs. The non-regrouped pigs had fewer skin lesions and better meat quality than the pigs regrouped at loading or in lairage. Pigs mixed at 80 kg at the farm had, in general, a comparable amount of skin lesions and comparable meat quality as the non-mixed group. If mixing is unavoidable, due to large within-group weight variations, mixing at 80 kg can be an alternative to reduce skin lesions at slaughter and to optimise meat quality.

## 1. Introduction

Concerning welfare, no regrouping of pigs is preferred. Regrouping of unfamiliar pigs is, however, a common management practice, which usually occurs not only at the transport procedure to the slaughterhouse, but at several stages of the pig’s life. In practice, it is also carried out at weaning, at the beginning of the growing-finishing period and within the groups of gestating sows. Aggressive encounters often result in skin lesions, altered behaviour, physiological effects, immunosuppressive effects [1] and reduced weight gain [2] and thus affect the profitability of pig production. Coutellier et al. [3] assume that the reduced weight is due to higher energy expenditure rather than to reduced feed intake. Every time pigs are regrouped, a new social order must be formed. This results in aggression in which pigs will fight intensely for approximately 2 h. Less intense fighting will continue to occur over the next couple of days until a relatively stable structure is formed [4]. 

The level of aggressive behaviour depends on several circumstances such as differences in age, gender, body weight, pen density, group size, the degree of familiarity, the behavioural composition of the group (active and passive copers), the circadian rhythm, the novelty of the environment, enrichment, whether food is provided ad libitum or restricted and genetics [1,5,6,7,8]. It has been reported that the heavier pigs in a group fight more and win more fights [6,8]. Ewbank and Bryant [5] reported that increased pen density leads to an increase in agonistic interactions. Group size may have an effect on how pigs react when they are mixed with unfamiliar individuals. Turner et al. [7] described that pigs from large groups (80 pigs per group) displayed a marked reduction in aggression compared with pigs from small groups (20 pigs per group). More recent studies have shown that aggressive behaviour after regrouping is moderately heritable and could be reduced by genetic selection [8,9]. 

In Belgium, 10.7 million pigs were slaughtered in 2019, correlating with the number of transported finisher pigs. The pig sector remains the most important supplier of carcasses, with 59% by weight. Annually, this amounts to 1.04 billion kg of carcass weight [10]. According to the most recent figures, more than 259 million pigs were slaughtered in the EU in 2018. The main suppliers were Germany, Spain, France and Poland [11].

In practice, unfamiliar pigs are frequently mixed prior transport to a slaughterhouse in order to obtain groups of uniform weight and to adjust the group size to the dimensions of the trailer compartments. Aggressive pig behaviour as a result of regrouping just before or after transport to the slaughterhouse induces pre-slaughter stress [9,12]. When unfamiliar conspecifics are mixed, pigs fight in order to determine a new dominance rank [13]. 

It is well known that in addition to skin lesions, stress caused by pre-slaughter regrouping also affects meat quality [14,15]. Regrouping of unfamiliar pigs should be avoided during the transport procedure. If mixing is unavoidable, then this must first take place at loading without regrouping in the further transport and slaughterhouse stages. According to van der Wal et al. [16], the aggressive behaviour of fattening pigs starts 10 to 20 minutes after arrival in the waiting pen of the slaughterhouse and continues until 140 to 210 min after arrival, after which the pigs become quiet and lie down. Once the new social hierarchy is established, this social hierarchy regulates aggressive interactions and reduces stressful social encounters in stable groups [17].

Aggression is a serious animal welfare concern which has led to many studies of methods to reduce the regrouping-associated aggression. Some examples have not been successful, like masking odours, sedatives and hide areas [18]. The purpose of this paper is twofold. Firstly, we want to investigate if the frequency of regrouping (at loading and in lairage) influences the incidence of skin lesions, which is an indicator of welfare and meat quality. Secondly, if regrouping is unavoidable, could mixing pigs in the last phase of fattening, at approximately 80 kg, optimise slaughter characteristics? In addition, is the incidence of the skin lesion score and meat quality of pigs mixed at 80 kg similar to the characteristics of the non-mixed pigs? Therefore, skin lesions and meat quality were determined in the slaughter line, and the comparison between carcasses of pigs mixed during the transport procedure and non-mixed pigs was made.

## 2. Materials and Methods

### 2.1. Animals and Housing

A total of 1332 hybrid pigs (Piétrain × Hypor), being heterozygous for the halothane gene, were used in this study. Both females and castrated males were raised at the Zootechnical Centre (ZTC, KU Leuven R&D, Lovenjoel, Belgium) per 12 pigs in mixed-sex groups. Pigs were housed in pens (9.53 m² per pen) with fully slatted floors and with a chain as point-source enrichment. After weaning, all pigs were individually marked with an ear tag number. All pigs had ad libitum access to water and received a commercial diet. The day before slaughter, all pigs were weighed individually using an electronic weighing scale and marked with an individual tattoo number on each side of the body. This number allows individual identification of carcasses in the slaughter line. Throughout this study, the same technicians took care of the pigs to exclude the effect of handling method on meat quality [19]. Animals were treated in accordance with the regulations of the Council Directive 86/609/EEC regarding the protection of animals used for experimental and other scientific purposes [20].

### 2.2. Grouping Strategies

The experiment focused on the transport of fattening pigs to the slaughterhouse. Therefore, pigs were assigned in mixed-gender groups of the following four grouping strategies:Uniform weight group formed at the start (22 kg) of the fattening. The unmixed strategy consisted of keepings pigs in the same social group during fattening, transport and lairage, called Mix 0;Uniform weight group formed at the start (22 kg) of the fattening and regrouped at loading the trailer at the farm, but without regrouping in the lairage (Mix 1);Uniform weight group formed at the start (22 kg) of the fattening, and regrouped at loading the trailer at the farm, and again at the lairage (Mix 2);Uniform weight group formed at the start (22 kg) of the fattening, regrouped at 80 kg at the farm, fattened till slaughter weight and transported to the slaughterhouse (without regrouping during the transport procedure) (Mix 80).

Pigs of grouping strategies Mix 0, Mix 2 and Mix 80 were never transported during the same transport. Each of these three grouping strategies was transported simultaneously with Mix 1. 

### 2.3. Transport

When the pigs weighed approximately 105 kg, they were transported by road from the Zootechnical Centre (Lovenjoel, Belgium) to the slaughterhouse (Comeco, Meer, Belgium). The distance from the Zootechnical Centre to the commercial slaughterhouse was 110 km. Pigs were driven from the pens to the trailer by the ZTC-technicians using lightweight driving boards. All pigs were loaded on the trailer by the truck driver using a tailgate lift. The pigs were transported in the upper four pens of a two-tiered trailer (with four pens per tier). Pigs were always transported before noon and per 12 in a trailer compartment (6.63 m² per compartment). The loading density did not exceed the maximum loading density (235 kg/m²) as determined in the EC Regulation [21] so that pigs were able to sit or lie down during transport. Pigs were fasted 16 hours before transport, but always had access to drinking water during housing at the ZTC. All transports were conducted by the same driver, the same trailer and the same truck. In addition, climate parameters were measured during each transport. Dry air temperature and humidity sensors (Miravox, Belgium) were fixed in the middle at 3 cm under the ceiling in each compartment of the trailer. Data on the wind velocity were provided by the Royal Meteorological Institute of Belgium (Brussels). Vents for natural ventilation were located alongside the whole trailer on both sides and on each tier. The THI is calculated by combining temperature and humidity using the method reported by Ravagnolo et al. [22]:THI = (1.8 × T + 32) − [(0.55 − 0.0055 × RH) × (1.8 × T − 26)]
where T = air temperature (°C) and RH = relative humidity (%). Transport time was defined as the time between departure from the farm and the arrival at the slaughterhouse. Slaughter dates were registered so that possible seasonal effects could be taken into account. The seasons wherein the pigs were transported to the slaughterhouse were defined as groupings of three whole months as identified by the Gregorian calendar: spring (21 March–20 June), summer (21 June–20 September), autumn (21 September–20 December) and winter (21 December–20 March).

### 2.4. Slaughterhouse

All pigs were transported to the same slaughterhouse over the same route. Each group of pigs was unloaded as soon as possible after the arrival at the slaughterhouse. The duration of unloading, defined as the time between the arrival at the slaughterhouse and the unloading, was always recorded. The lairage was equipped with a showering system (which was used) and drinking nipples. The pen surface was 6.14 m^2^. The time between unloading and the moment pigs were driven to the stunning area is defined as lairage time, and was recorded. Head-to-back electrical stunning (240 V, 800 Hz for 2 s), which induces cardiac arrest, was applied. 

### 2.5. Skin Lesions

The skin lesion score is associated with aggressive behaviour and, therefore, useful to analyse the fighting behaviour of pigs [23]. The skin of the left carcass side of the transported pigs was evaluated in the slaughter line 30 min post-mortem by the same team (2 persons) throughout 30 transports. After scalding and evisceration, skin lesions were visually assessed in different parts of the left carcass, i.e., shoulder, middle and ham, using the anatomical locations and procedures described by Barton Gade et al. [24]. A 4-point photographic scale was used, starting at 1 = no damage and 4 = extreme damage (Figure 1). The used scale can be considered as a combination of the number and the severity of the lesions. Only recent/fresh skin lesions were registered. The change in colour from red (recent lesion) to yellow (older lesion) offers the possibility to differentiate between recent and older skin lesions [25]. Fresh skin lesions may indicate damage due to transport and lairage conditions [26].

### 2.6. Meat Quality

In each lairage compartment, six pigs were selected at random to evaluate meat quality after slaughter, but, during some evaluation moments, one pig was missed and only five pigs per lairage compartment were effectively measured. The left carcass side was used for all measurements. Thirty-five minutes post-mortem, carcasses were graded with an SKGII-device (Schlachtkörper Klassifizierungs Gerät, Tecpro GmbH, Germany), which combines four physical measurements (ham angle, ham width, loin width and back fat thickness) to estimate the lean meat content [27]. The carcass was weighed at the same time. In the *m. longissimus dorsi*, at a depth of approximately 7 cm between the fourth and the fifth back rib, pH (pH_1_) and temperature were determined 45-min post-mortem. A pH-meter equipped with an insertion glass electrode and a temperature probe (PH/PT-STAR, R. Matthäus, Pöttmes, Germany) was used. Calibration uses standard buffers at pH 4 and 7 [28]. The pH electrode was cleaned at the start and after every 10 measurements with a cleaning solution for oils. The electrode reading was checked after cleaning with standard solutions of pH 4 and 7.

After chilling at 2 °C, carcasses were commercially cut and transported to a grocery store where the meat quality of the loin (*m. longissimus dorsi*) was measured 48 h post-mortem. Electrical conductivity (EC) with the PQM (Pork Quality Meter, Intek Klassifizierungs-technik, Aichach, Germany) and pH_u_ (PH / PT-STAR) were measured in the loin between the fourth and the fifth back ribs of the *m. longissimus dorsi* before cutting. The colour of the loin in the transverse cut between the fourth and fifth (back) rib was measured using the Commission Internationale de l’Eclairage (1976) (CIE) values (L*, a* and b*) with a chromameter (CR300, Minolta, Osaka, Japan). The L* value represents lightness where L* = 0 is completely black, and L* = 100 is completely white. The a* value represents red-green colours: positive a* values mean red colours and negative a* values mean green colours. The b* value represents yellow-blue colours: positive b* values mean yellow colours and negative b* values mean blue colours [29]. Each instrument was calibrated following the manufacturer’s instructions before each use. In addition, the Japanese colour standard (JCS) was used to evaluate meat colour (1 = pale gray to 6 = dark purple). The Japanese Colour Grades 1 and 2 are related to PSE meat, and Colour Grades 3 and 4 to normal meat [30]. On this cut, drip losses (DRIP) were determined with the filter paper method [31]. All meat quality parameters were always measured by the same team (2 persons).

### 2.7. Statistical Analyses

Pigs were considered as the experimental units because data on skin lesions and meat quality were collected for each individual pig. Statistical significance was accepted at *p* < 0.05, tendencies were accepted at 0.05 < *p* < 0.1. The normality and linearity of the dependent variables were determined before statistical analysis (UNIVARIATE procedure). Pearson correlation coefficients were calculated between carcass conformation variables and meat quality variables. Ham angle had the highest correlation with the meat quality parameters, and hence, ham angle as carcass conformation parameter is included in the used model. 

Skin lesion data were analysed with the GLIMMIX procedure of SAS 9.4 [32]. The applied procedure (GLIMMIX) makes it possible to allocate a random effect (i.e., transport number) to a variable so that the pigs can be regarded as the experimental units. Per carcass location, namely shoulder (gender, wind velocity and ham angle), middle (gender, wind velocity and lairage) and ham (wind velocity and lairage) independent variables were introduced as covariates in the model to analyse the effect of the grouping strategy. These parameters were included in the model if *p* < 0.05. Furthermore, frequencies of skin lesions per grouping strategy were calculated.

Meat quality data were analysed with the Mixed procedure of SAS 9.4. A model was built per meat quality parameter (carcass temperature, pH_1_, pH_u_, EC, DRIP, JCS, L, a, b) with transport number as a random factor and the independent variables (gender, transport season, THI, wind velocity, and ham angle).

## 3. Results

Spread over 30 transports, 1332 pigs were transported to the slaughterhouse. Skin lesions and meat quality of, respectively, 1314 and 620 carcasses were registered (Table 1). No animals were transported in the autumn. The mean transport, unloading and lairage time were, respectively, 102 ± 17 min (maximum: 160; minimum: 85), 25 ± 13 min (maximum: 56; minimum: 8) and 98 ± 27 min (maximum: 120; minimum: 40). 

### 3.1. Skin Lesions

The grouping strategy influenced the incidence of skin lesions on shoulder (*p* < 0.001), middle (*p* < 0.001) and ham (*p* = 0.006). Mix 0 and Mix 80 showed less damage, both at the level of the shoulder, the middle and the ham, than Mix 1 and Mix 2 (Table 2). 

### 3.2. Meat Quality

As shown in Table 3, the carcass temperature was influenced by the grouping strategy. The carcass temperature of Mix 0 and Mix 80 is lower than Mix 1 and Mix 2. According to Table 3, the EC of Mix 0 and Mix 80 is lower than EC of Mix 1. Mix 80 had a lower JCS than Mix 1 and Mix 2. In addition, Mix 80 has a higher L* value than Mix 1 and Mix 2.

There was also a tendency (*p* < 0.1) that DRIP (*p* = 0.058) is influenced by grouping strategy. It seems that carcasses of Mix 1 has a higher drip loss than carcasses from Mix 0 and Mix 80. The grouping strategies did not influence pH_1_ (*p* = 0.803) 45 min post-mortem, nor pH_u_ (*p* = 0.682), a* value (*p* = 0.188) and b* value (*p* = 0.615). 

## 4. Discussion

Grouping has an impact on the severity and incidence of skin lesions, but also on meat quality. However, pigs that were never regrouped still had recent skin lesions on the shoulder (15.7%), middle (19.1%) and ham (6.4%), caused by the transport procedure. Looking at the shoulder damage, it can be concluded that, although pigs are not regrouped, transporting pigs is still accompanied by a certain level of aggression by one-way bites [33,34]. During transport, some pigs may have difficulties in standing up, lose balance and fall. In such situations, pen mates can tramp the fallen pig, resulting in skin lesions due to mounting [35]. Moving animals to a new environment, in this particular case, the trailer and the lairage, may provoke some aggression even among familiar animals [34], possibly because of the effect of stress on social memory, i.e., the ability to recognise (recent) groupmates [36]. It has been suggested that because of the lack of suitable material for exploration, pigs redirect their explorative behaviour to the pen mates [37,38,39]. Indeed, Peeters and Geers [40] detected lower skin damage using enrichment in the trailer and lairage. However, the impact on meat quality was minimal; only an effect on pH_1_ was seen.

The carcass temperature, EC and DRIP were shown to be better for non-regrouped pigs (Mix 0), and the pigs regrouped at 80 kg at the farm (Mix 80). On the other hand, the colour of Mix 80, determined by the JCS, seemed to be paler. However, the colour of Mix 80 was still within the range of acceptable colour characteristics [30,41]. On the other hand, the L* value, which is an indicator for colour lightness, between Mix 0 and Mix 80 did not differ. Dalla Costa et al. [35] explains the discrepancy between the JCS and L* value by the confounding effect of the light reflectance of marbling fat. Because of fat colour reflectance, higher marbling scores may have resulted in higher L* values, while visual colour scores are not influenced by marbling scores.

It is well known that short term stress in lairage influences pork quality. The quality of handling pigs in lairage influences meat quality [42,43]. Furthermore, the lairage environment (water cooling, the level of noise, pig density, group size, air temperature, relative humidity, ventilation) during the resting period influences the pigs [44]. However, it seems that double regrouping does not have a higher impact on skin lesions and meat quality than regrouping only once. This might be due to the fact that the animals cannot establish a new social rank in a moving trailer and begin to fight in the lairage to set a new rank [45]. The suggestion is supposed by the research of Guise and Penny [46] that regrouping pigs at any stage of transportation will result in fighting and skin damage. 

Our findings confirm earlier research [46], pointing out that regrouping pigs increases skin lesions and reduces meat quality. Therefore, the installation of mobile dividing gates in the trailer, but also in the slaughterhouse, makes it possible to adjust the compartment space to the group size and makes it more facile to keep the group together from farm till slaughterhouse without fighting to establish a new social hierarchy [25].

The results of our study suggest that there are benefits of forming uniform weight groups at the start of the finishing period and excluding regrouping in the transport procedure in terms of reducing skin lesions at slaughter and optimising meat quality. In addition, regrouping at the last fattening stage has no long-term adverse effects on these skin and meat parameters. However, aggressive behaviour was significantly elevated at regrouping [47]. While pigs were standardized in weight (80 kg) at regrouping, there was a large variation in live weight (weight one day before slaughtering: from 84.0 to 112.5 kg). Economically, regrouping pigs at 80 kg is not interesting. Therefore regrouping in the finishing stage is questionable and should only be carried out if there are no other options.

Non-regrouping of pigs is the best option concerning animal welfare and meat quality. However, regrouping of finisher pigs is just done in order to transport pig batches, uniform in weight, to the slaughterhouse. In fact, the focus should be more on limiting weight differences during rearing. However, this is not self-evident as many parameters, e.g., genetics, weaning age, housing, animal health, and hierarchal order, can affect growth and final weight [48,49,50,51,52]. 

## 5. Conclusions

Often, in general pig husbandry, unfamiliar pigs are mixed just before transport to the slaughterhouse, due to unequal growth rate in a pen. No regrouping of pigs during the fattening, transport and slaughterhouse stages results in a lower incidence of skin lesions at slaughter and better meat quality. If mixing is unavoidable, due to the large within-group variations in weight, mixing at 80 kg (and kept together till slaughter) can be an alternative to reduce skin lesions at slaughter and to optimise meat quality. Concerning carcass damage and meat quality, mixing pigs one (before transport) or two times (before transport and in lairage) does not differ.

## Figures and Tables

**Figure 1 animals-10-00544-f001:**
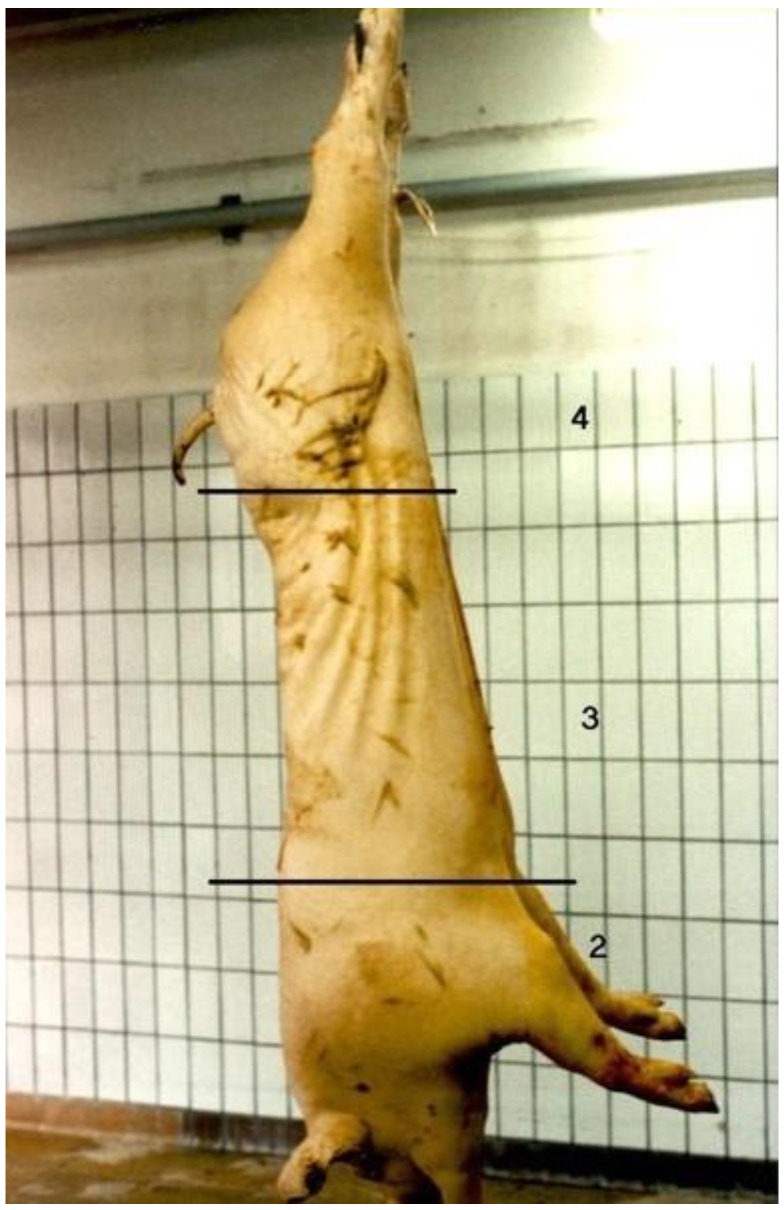
Scoring skin lesions in different parts, i.e., shoulder, middle and ham, of the left carcass, and examples of skin lesions in scores 2 to 4 on the scale of 1 = no damage and 4 = extreme damage [24].

**Table 1 animals-10-00544-t001:** Grouping strategy with Mix 0: pigs kept together in the same social group composition during fattening, transport and lairage; Mix 1: pigs only regrouped at loading; Mix 2: pigs regrouped at loading and in lairage; Mix 80: pigs regrouped at approximately 80 kg and further kept together at farm, transport and lairage. No animals were transported in autumn. Animals/population: number of animals per total number of transported animals (N = 1332). Skin lesions: carcasses checked for skin lesions. Meat quality: carcasses for which meat quality is determined.

Grouping Strategy	Category	Winter	Spring	Summer	Total
Mix 0	Animals (n)	84	72	84	240
	Animals/population (%)	6.3	5.4	6.3	18.0
	Pens (n)	7	6	7	20
	Transports (n)	5	3	4	12
	Skin lesions (n)	82	72	82	236
	Meat quality (n)	35	29	36	100
Mix 1	Animals (n)	276	228	240	744
	Animals/population (%)	20.7	17.1	18.1	55.9
	Pens (n)	23	19	20	62
	Transports (n)	11	9	10	30
	Skin lesions (n)	273	226	236	735
	Meat quality (n)	115	95	100	310
Mix 2	Animals (n)	72	72	60	204
	Animals/population (%)	5.4	5.4	4.5	15.3
	Pens (n)	6	6	5	17
	Transports (n)	3	3	3	9
	Skin lesions (n)	71	72	59	202
	Meat quality (n)	36	38	27	101
Mix 80	Animals (n)	36	48	60	144
	Animals/population (%)	2.7	3.6	4.5	10.8
	Pens (n)	3	4	5	12
	Transports (n)	3	3	3	9
	Skin lesions (n)	36	46	59	141
	Meat quality (n)	29	37	43	109

**Table 2 animals-10-00544-t002:** Frequencies of skin lesions per score and per grouping strategy.

Grouping Strategy ^a^	Skin Lesion Score ^b^	Shoulder Damage (n–%)	Middle Damage (n–%)	HamDamage (n–%)
Mix 0	1	199–84.3	191–80.9	221–93.6
	2	25–10.6	40–17.0	15–6.4
	3	11–4.7	4–1.7	0–0.0
	4	1–0.4	1–0.4	0–0.0
Mix 1	1	502–68.3	515–70.1	648–88.2
	2	158–21.5	175–23.8	69–9.4
	3	60–8.2	40–5.4	15–2.0
	4	15–2.0	5–0.7	3–0.4
Mix 2	1	132–65.3	145–71.8	175–86.6
	2	45–22.3	45–22.3	20–9.9
	3	15–7.4	10–5.0	6–3.0
	4	10–5.0	2–1.0	1–0.5
Mix 80	1	119–84.4	113–80.1	130–92.2
	2	15–10.6	24–17.0	10–7.1
	3	6–4.3	4–2.8	1–0.7
	4	1–0.7	0–0.0	0–0.0

^a^ Grouping strategy with Mix 0: pigs kept together in the same social group during fattening, transport and lairage; Mix 1: pigs only regrouped at loading; Mix 2: pigs regrouped at loading and in lairage; Mix 80: pigs regrouped at approximately 80 kg and further kept together at farm, transport and lairage. ^b^ Skin lesion score: 4-point scale with 1 = no damage and 4 = extreme damage.

**Table 3 animals-10-00544-t003:** Effect of grouping strategy on meat quality parameters of the *m. longissimus dorsi* muscle 45 min and 48 h post-mortem. Data are presented as Ls means ± standard error. Temperature and pH_1_ were determined 45 min post-mortem, the other values after 48 h post-mortem.

	*p*	Grouping Strategies			
		Mix 0	Mix 1	Mix 2	Mix 80
Carcass temperature, °C	=0.008 *	36.9 ± 0.24 ^a^	37.5 ± 0.18 ^b^	37.5 ± 0.27 ^b^	36.9 ± 0.24 ^a^
pH_1_	=0.803	6.05 ± 0.03	6.04 ± 0.02	6.01 ± 0.04	6.06 ± 0.04
pH_u_	=0.682	5.52 ± 0.02	5.51 ± 0.02	5.52 ± 0.02	5.54 ± 0.05
EC, μs	<0.001 *	6.48 ± 0.28 ^a^	6.99 ± 0.24 ^b^	6.61 ± 0.32 ^ab^	6.31 ± 0.29 ^a^
DRIP, mg fluid	=0.058 *	58.7 ± 4.12 ^a^	64.5 ± 2.53 ^b^	67.0 ± 4.38 ^ab^	52.0 ± 5.20 ^a^
JCS	<0.001 *	2.71 ± 0.05 ^ab^	2.81 ± 0.03 ^a^	2.81 ± 0.06 ^a^	2.59 ± 0.05 ^b^
L*	=0.015 *	54.6 ± 0.54 ^ab^	53.8 ± 0.34 ^a^	53.2 ± 0.92 ^a^	55.5 ± 0.45 ^b^
a*	=0.188	7.56 ± 0.39	7.18 ± 0.27	6.93 ± 0.40	6.82 ± 0.35
b*	=0.615	4.98 ± 0.32	5.24 ± 0.17	5.27 ± 0.25	5.08 ± 0.31

^a,b^ Ls means and standard errors without a common superscript letter differ. * Statistically significant difference (GLIMMIX, *p* < 0.05).

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
