# Peer review of "The Impact of Grouping on Skin Lesions and Meat Quality of Pig Carcasses"

_animals, 2020, doi:10.3390/ani10040544_

Round 1

Reviewer 1 Report

The article is well written, so it is easy to follow.

I have some minor recommendations:

Line 59: aggression towards pigs from other groups
.....Q.: which other groups? Less than 80 kg? Please, rephrase.

Figure 1. I believe that Y axis values must be with "." instead of ","
The Figure also looks light grey, could you improve the quality or maybe put it in black format.
I have also not seen Figure 1 being called in the text before it shows up.

I felt like more discussion is needed. For instance..lines 255-257...what's the implication of paler meat? PSE? what are the consequences?
Line 456 - Write after: colour of Mix 80 seems to be paler, however it does not differ from Mix 0.
Also, Japanese colour scale parameters is not discussed and neither associated to Chromameter measurements.

There is a manuscript published in Animal Journal that may help in the discussion of meat quality, pH, PSE and so on:
Dalla Costa, F. A., Lopes, L. S., & Dalla Costa, O. A. (2017). Effects of the truck suspension system on animal welfare, carcass and meat quality traits in pigs. Animals, 7(5), 1–13.

Another one about skin lesions and transport:
Dalla Costa, F. A., Paranhos da Costa, M. J. R., Faucitano, L., Dalla Costa, O. A., Lopes, L. S., & Renuncio, E. (2016). Ease of handling, physiological response, skin lesions and meat quality in pigs transported in two truck types. Archivos de Medicina Veterinaria, 48(3), 299–304.

Author Response

Response to Reviewer 1 Comments

Point 1: The article is well written, so it is easy to follow.

Respons 1: The authors thank the reviewer for thoroughly reading the manuscript and providing suggestions that improve the manuscript.  

Point 2: Line 59: aggression towards pigs from other groups
.....Q.: which other groups? Less than 80 kg? Please, rephrase.

Respons 2: Indeed, the sentence was not correct, we changed it in: Turner et al. [7] described that pigs from large groups (80 pigs per group) displayed a marked reduction in aggression compared with pigs from small groups (20 pigs per group).

Point 3: Figure 1. I believe that Y axis values must be with "." instead of ","
The Figure also looks light grey, could you improve the quality or maybe put it in black format. I have also not seen Figure 1 being called in the text before it shows up.

Respons 3: Indeed, the figure could be better. However, based on your comments and from the other reviewers, we decided to delete the Figure 1 because it add no more information than Table 3.

Point 4: I felt like more discussion is needed. For instance..lines 255-257...what's the implication of paler meat? PSE? what are the consequences?

Respons 4: There were 2 reasons why we chose to present it this way: 1) the focus is on the grouping strategy, not on PSE. 2) we determined PSE indicators, but we did not determine PSE itself (this should be done by histological research). The values of the indicators depend on numerous factors, such as genetics (we used extreme Pietrain crossings), environmental factors ... This means that certain values are considered normal in one continent/country, while elsewhere they are considered abnormal (PSE, DFD...). Hopefully the reviewer will agree with our view.

Point 5: Line 456 - Write after: colour of Mix 80 seems to be paler, however it does not differ from Mix 0.

Respons 5: The original sentence was confusing (it was not clear if we mentioned the JCS or the L* value), so we changed the sentence in: On the other hand, the colour of Mix 80, determined by the JCS, seems to be paler.

Point 6: Also, Japanese colour scale parameters is not discussed and neither associated to Chromameter measurements.

Respons 6: Thank you for drawing our attention to that shortcoming. We have adjusted this and discussed the discrepancy between the JCS and the L* value by the confounding effect of the light reflectance of marbling fat.

Point 7: There is a manuscript published in Animal Journal that may help in the discussion of meat quality, pH, PSE and so on.

Respons 7: Thank you, we got some interesting information from the paper, see line 288-290 (the falling of animals) and line 300-304 (the confounding effect of the light reflectance of marbling fat).

Reviewer 2 Report

Dear Authors,

The manuscript ‘The impact of grouping strategies on skin lesions and meat quality of pig carcasses’ provides novel, important information regarding the impacts of re-grouping of pigs during transport. Millions of pigs are transported annually, and it is well-known that transport is a major cause of stress. Therefore, it is important that studies focusing on the methods aiming at reducing the stress are conducted. In general, the paper is well-written and easy to read; however, there are some parts of the text that could be clearer, please see specific comments below.

Headline

Lines 2-3: Could the headline be shortened by removing the word ‘strategies’?

Simple summary

Line 15: Please replace ‘In a study’ with ‘In this study’

Line 15: Could the word ‘treatments’ be replaced with ‘grouping strategies, namely…’? Check throughout the text.

Line 16: Please replace ‘were compare. Therefore…’ by ‘were compared by determining skin lesions and meat quality at slaughter.’

Line 22: And probably also with high weight variations at slaughter.

Abstract

Line 30: Please replace ‘treatment showed’ with ‘pigs had’

Line 31: Please consider replacing ‘the groups regrouped’ with ‘regrouped pigs’  

Keywords

Please consider adding a word ‘transport’ in keywords

Introduction

The Introduction section is in general well-written and gives a reader a basic knowledge about the impacts of re-grouping of pigs. Maybe you could start the introduction with the paragraph starting from the line 62 (followed by the paragraph from line 51) giving first basic information about regrouping and after that go more in details. It would also be interesting to see some figures about the numbers of pigs transported to slaughterhouse (e.g. in Belgium or in Europe).  

Lines 41-42:  Please reword the sentence ‘These fights occur when unfamiliar conspecifics are mixed in order to determine a new dominance rank.’ as now the reader gets an impression that animals are mixed to achieve a new dominance rank.

Line 76: You mention the impact of ‘the frequency of regrouping’; did you perform pair-wise comparisons? I would like to see more details concerning the pair-wise comparison in the Material and Methods section. Incidence or severity (see comments below)?

Line 81: Please consider deleting the word ‘with’.

Material and Methods

Line 87: What was the size of a pen?

Line 99: Can the word ‘treatment’ be replaced with ‘grouping strategies’?

Line 106: Please add ‘at the farm’

Line 120: What was the size of a trailer compartment?

Line 142: What was the size of a lairage compartment?

Line 153: Could you add a photograph example of skin lesions as a supplementary material (e.g. one picture per scale)? However, this is not required if there are photo examples in the article you referred at (I wasn’t able find the original reference).

Line 159: Please specify the selection of pigs (e.g. how many pigs per a group)

Line 176-177: Please add the average values or normal range for L*, a* and b* if possible

Line 185 Statistical analysis are not clear. In general, please be more specific while defining the used analysis. With the present data it would not be possible to re-conduct the study.

Line 186: I guess also the data on skin lesions were collected for each individual animal? Should you mention this?

Line 188: By which statistical method?

Line 189: Please clarify the sentence: ‘Ham angle had the highest correlation with the meat quality parameters’

Lines 193-194: Please clarify ‘Per carcass location, namely shoulder (gender, wind velocity and ham angle), middle (gender, wind velocity and lairage) and ham (wind velocity and lairage) independent variables were introduced as covariates in the model to analyse the effect of the grouping strategy.’ Why the gender, wind velocity and ham angle/lairage were selected?

Results

The idea of presenting most of the results in Tables and a Figure is excellent. This makes the text easy to read and the most important results are easily found. Good! However, the are some changes that I want to suggest.

Lines 206-207: Please consider not using seconds. I think it is enough to present only total minutes.

Table 1: I would prefer presenting the number of animals in the first row, the number of groups in the second row and transports in the third row. Please also add the proportion of pigs from the whole study population (e.g. below the number of animals). Moreover, I would add ‘pigs/carcasses examined/checked/assessed for skin lesions’ and same for the meat quality.

Table 2. Did you assess whether there were any lesions or not (if yes, then the severity) or did you also count the total number of lesions per carcass? This could be better explained in the Material and Methods section. Please add the n of the damages, e.g. shoulder damage % (n) and please consider using only one decimal.   

Line 228: Should you use ‘was influenced by’ instead of ‘influences’?

Table 3. Please use maximum of three decimals for p-values (same applies throughout the text). Please better indicate what measures were taken after 45 minutes and which ones after 48 hours and what the abbreviations mean (temperature and pH are clear to readers, but the others might not be). What does the upper letters of a, b and ab mean? Please highlight the statistically significant p-values with some upper letter.

Figure 1. I wonder why to represent the same results as in Table 3. Consider deleting the Figure 1. If not, please replace commas with dots. Please clarify ‘Within an experiment, means without a common letter differ, p < 0.05. (ie. explain the meaning for the upper letters betters). Consider explaining L* and EC.

Discussion

Line 245: Please start with the most important findings, ie. the grouping has impact on the severity (and incidence) of skin lesions and on meat quality.

Line 250: Please clarify ‘social memory’

Lines 255-257: Please consider replacing ‘is better for’ with ‘was shown to be better for’ and do not use abbreviations, such as Mix O, but explain them

Lines 261-262: Please check if this is correct ‘However, it seems that regrouping pigs at loading and in lairage does not exacerbate the incidence of skin lesions and does not diminish the meat quality.’ I think you mean that double re-grouping does not have a higher impact than re-grouping only once; please clarify (e.g. adding a word ‘both’)

Line 267: Maybe some discussion about the weight differences among pigs could be added?

Conclusion

Line 285: Why do you mention a drip loss (which is a specific measure) and a carcass damage (which is a general measure)? Can you just say meat quality (or where the statistically significant differences between the two strategies in some values?) and carcass damage?

References

Without doi's the references were not easily found; therefore, not all the references were checked. 

Author Response

Point 1: The manuscript ‘The impact of grouping strategies on skin lesions and meat quality of pig carcasses’ provides novel, important information regarding the impacts of re-grouping of pigs during transport. Millions of pigs are transported annually, and it is well-known that transport is a major cause of stress. Therefore, it is important that studies focusing on the methods aiming at reducing the stress are conducted. In general, the paper is well-written and easy to read; however, there are some parts of the text that could be clearer, please see specific comments below.

Respons 1: The authors thank the reviewer for the meticulous reading and the constructive suggestions that take the article to a higher level.

Point 2: Lines 2-3: Could the headline be shortened by removing the word ‘strategies’?

Respons 2: Okay, we removed ‘strategies’ in the headline.

Point 3: Line 15: Please replace ‘In a study’ with ‘In this study’

Respons 3: Okay, we did that.

Point 4: Line 15: Could the word ‘treatments’ be replaced with ‘grouping strategies, namely…’? Check throughout the text.

Respons 4: Throughout the text we deleted the word ‘treatments’ and replaced it.

Point 5: Line 16: Please replace ‘were compare. Therefore…’ by ‘were compared by determining skin lesions and meat quality at slaughter.’

Respons 5: we replaced it.

Point 6: Line 22: And probably also with high weight variations at slaughter.

Respons 6: we added the suggestion.

Point 7: Line 30: Please replace ‘treatment showed’ with ‘pigs had’

Respons 7: the adapted sentence: The non-regrouped pigs had less skin lesions and…

Point 8: Line 31: Please consider replacing ‘the groups regrouped’ with ‘regrouped pigs’  

Respons 8: we replaced it.

Point 9: Please consider adding a word ‘transport’ in keywords

Respons 9: we added ‘transport’.

Point 10: The Introduction section is in general well-written and gives a reader a basic knowledge about the impacts of re-grouping of pigs. Maybe you could start the introduction with the paragraph starting from the line 62 (followed by the paragraph from line 51) giving first basic information about regrouping and after that go more in details. It would also be interesting to see some figures about the numbers of pigs transported to slaughterhouse (e.g. in Belgium or in Europe).  

Respons 10: we put some information about it in the manuscript: There is no detailed information about the number of the transported finisher pigs in Belgium, but the number of slaughtered pigs is known, what is correlated with the number of transported pigs. In Belgium 10.7 million pigs were slaughtered in 2019. Compared with 2018, it is a decrease of 5%. The pig sector remains the most important supplier of carcasses with 59% by weight. Annually, this amounts to 1.04 billion kg carcass weight [10]. According to the most recent figures, more than 259 million pigs were slaughtered in the EU in 2018. The main suppliers were Germany, Spain, France and Poland [11].

Point 11: Lines 41-42:  Please reword the sentence ‘These fights occur when unfamiliar conspecifics are mixed in order to determine a new dominance rank.’ as now the reader gets an impression that animals are mixed to achieve a new dominance rank.

Respons 11: we reworded the sentence: When unfamiliar conspecifics are mixed, pigs fight in order to determine a new dominance rank [13].

Point 12: Line 76: You mention the impact of ‘the frequency of regrouping’; did you perform pair-wise comparisons? I would like to see more details concerning the pair-wise comparison in the Material and Methods section. Incidence or severity (see comments below)?

Respons 12: incidence or severity: we used a combination of it. Therefore we added a figure, like you suggested.  

Point 13: Line 81: Please consider deleting the word ‘with’.

Respons 13: we deleted the word.

Point14: Line 87: What was the size of a pen?

Respons 14: 9.53 m² per pen.

Point 15: Line 99: Can the word ‘treatment’ be replaced with ‘grouping strategies’?

Respons 15: we did that.

Point 16: Line 106: Please add ‘at the farm’

Respons 16: we did that.

Point 17: Line 120: What was the size of a trailer compartment?

Respons 17: 6.63 m² per trailer compartment.

Point 18: Line 142: What was the size of a lairage compartment?

Respons 18: 6.14 m² per pen.

Point 19: Line 153: Could you add a photograph example of skin lesions as a supplementary material (e.g. one picture per scale)? However, this is not required if there are photo examples in the article you referred at (I wasn’t able find the original reference).

Respons 19: we added an own photograph with the different scores.

Point 20: Line 159: Please specify the selection of pigs (e.g. how many pigs per a group)

Respons 20: we reworded it: In each lairage compartment, 6 pigs were selected at random to evaluate meat quality after slaughter, but during some evaluation moments 1 pig was missed and only 5 pigs per lairage compartment were effectively measured.

Point 21: Line 176-177: Please add the average values or normal range for L*, a* and b* if possible

Respons 21: we added this information: The L* value represents lightness where L* = 0 is completely black, and L* = 100 is completely white. The a* value represents red-green colours: positive a* values mean red colours and negative a* values mean green colours. The b* value represents yellow-blue colours: positive b* values mean yellow colours and negative b* values mean blue colours [29].

Point 22: Line 186: I guess also the data on skin lesions were collected for each individual animal? Should you mention this?

Respons 22: Indeed, we corrected it.

Point 23: Line 188: By which statistical method?

Respons 23: we used the proc univariate.

Point 24: Line 189: Please clarify the sentence: ‘Ham angle had the highest correlation with the meat quality parameters’

Respons 24: We added sentence before the sentence you mentioned:  Pearson correlation coefficients were calculated between carcass conformation variables and meat quality variables. Ham angle had the highest correlation with the meat quality parameters and hence ham angle as carcass conformation parameter is included in the used model.

Point 25: Lines 193-194: Please clarify ‘Per carcass location, namely shoulder (gender, wind velocity and ham angle), middle (gender, wind velocity and lairage) and ham (wind velocity and lairage) independent variables were introduced as covariates in the model to analyse the effect of the grouping strategy.’ Why the gender, wind velocity and ham angle/lairage were selected?

Respons 25: we included this sentence: These parameters were included in the model if p < 0.05.

Point 26: Lines 206-207: Please consider not using seconds. I think it is enough to present only total minutes.

Respons 26: We adjusted it like you suggested.

Point 27: Table 1: I would prefer presenting the number of animals in the first row, the number of groups in the second row and transports in the third row. Please also add the proportion of pigs from the whole study population (e.g. below the number of animals). Moreover, I would add ‘pigs/carcasses examined/checked/assessed for skin lesions’ and same for the meat quality.

Respons 27: we adjusted Table 1.

Point 28: Table 2. Did you assess whether there were any lesions or not (if yes, then the severity) or did you also count the total number of lesions per carcass? This could be better explained in the Material and Methods section. Please add the n of the damages, e.g. shoulder damage % (n) and please consider using only one decimal.  

Respons 28: No, we did not count the number of lesions, we just gave it a score from 1 till 4 (see also figure 1). We put the n in the table. 

Point 29: Line 228: Should you use ‘was influenced by’ instead of ‘influences’?

Respons 29: okay, we replaced it.

Point 30: Table 3. Please use maximum of three decimals for p-values (same applies throughout the text). Please better indicate what measures were taken after 45 minutes and which ones after 48 hours and what the abbreviations mean (temperature and pH are clear to readers, but the others might not be). What does the upper letters of a, b and ab mean? Please highlight the statistically significant p-values with some upper letter.

Respons 30: we put extra information in the header of Table 3: Temperature and pH1 were determined 45 min post-mortem. Determined 48h post-mortem: electrical conductivity (EC), pHu, drip loss (DRIP), the Japanese colour standard (JCS) and the L*, a* and b* values by using a chromameter. a ,bWithin a column and variable, Ls Means and standard errors without a common superscript letter differ. *p < 0.05, **p < 0.05, ***p < 0.001.

Point 31: Figure 1. I wonder why to represent the same results as in Table 3. Consider deleting the Figure 1. If not, please replace commas with dots. Please clarify ‘Within an experiment, means without a common letter differ, p < 0.05. (ie. explain the meaning for the upper letters betters). Consider explaining L* and EC.

Respons 31: as suggested, we deleted Figure 1.

Point 32: Line 250: Please clarify ‘social memory’

Respons 32: it is a term used by Held et al. According to them, it is the ability to recognize (recent) groupmates.

Point 33: Lines 255-257: Please consider replacing ‘is better for’ with ‘was shown to be better for’ and do not use abbreviations, such as Mix O, but explain them

Respons 33: Okay, we did that.

Point 34: Lines 261-262: Please check if this is correct ‘However, it seems that regrouping pigs at loading and in lairage does not exacerbate the incidence of skin lesions and does not diminish the meat quality.’ I think you mean that double re-grouping does not have a higher impact than re-grouping only once; please clarify (e.g. adding a word ‘both’)

Respons 34: indeed, therefore we replaced the sentence: However, it seems that double regrouping does not have a higher impact on skin lesions and meat quality than regrouping only once.

Point 35: Line 267: Maybe some discussion about the weight differences among pigs could be added?

Respons 35: we added this: Non-regrouping of pigs is the best option in function of skin lesions and meat quality. However, regrouping of finisher pigs is just done in order to transport pig batches, uniform in weight, to the slaughterhouse. In fact, the focus should be more on limiting weight differences during rearing. However, this is not self-evident as many parameters, e.g. genetics, weaning age, housing, animal health, and hierarchal order, can affect growth and final weight [48-52].

Point 36: Line 285: Why do you mention a drip loss (which is a specific measure) and a carcass damage (which is a general measure)? Can you just say meat quality (or where the statistically significant differences between the two strategies in some values?) and carcass damage?

Respons 36: indeed, we adjusted it: Concerning carcass damage and meat quality, mixing pigs 1 (before transport) or 2 times (before transport and in lairage) does not differ.  

Point 37: Without doi's the references were not easily found; therefore, not all the references were checked. 

Respons 37: the doi’s were added.

Reviewer 3 Report

The authors present a nice study on the effect of mixing on carcass quality of pigs. The topic is of high relevance for the pork chain, as stressful conditions influence carcass quality up to meat quality and off odors of pork. The results clearly show, that a mixing during the late fattening period is able to avoid negative effects of mixing for carcass quality. The parameter are appropriate and the results convincing. A very nice and sound study which provides a solution for meat quality problems due to mixing of groups before slaughter and should be published.

Author Response

The authors present a nice study on the effect of mixing on carcass quality of pigs. The topic is of high relevance for the pork chain, as stressful conditions influence carcass quality up to meat quality and off odors of pork. The results clearly show, that a mixing during the late fattening period is able to avoid negative effects of mixing for carcass quality. The parameter are appropriate and the results convincing. A very nice and sound study which provides a solution for meat quality problems due to mixing of groups before slaughter and should be published.

Respons: The authors thank the reviewer for the positive comments and the appreciation.

Round 2

Reviewer 2 Report

Dear Authors,

I perceive that the manuscript has improved a lot. I only have some minor suggestions left. See below 

Line 19: Please replace the word “groups” with “pigs”

Line 32: Please replace the word “groups” with “pigs”

Line 36: Please put the keywords in alphabetical order

Lines 61-67: Please consider compressing the sentence into “In Belgium, 10.7 million pigs were slaughtered in 2019, correlating with the number of transported finisher pigs. The pig sector remains the most important supplier of carcasses with 59% by weight. Annually, this amounts to 1.04 billion kg carcass weight [10]. According to the most recent figures, more than 259 million pigs were slaughtered in the EU in 2018. The main suppliers were Germany, Spain, France and Poland [11].”

Line 68: Please replace “prior to loading” to “prior transport to a slaughterhouse”

Line 92: Please change to “concern which has led” 

Figure 1: This is an excellent photo indicating how the skin lesions were scored. Could the caption be: “Scoring skin lesions in different parts, i.e. shoulder, middle and ham, of the left carcass, and examples of skin lesions in scores 2 to 4 in scale of 1 = no damage and 4 = extreme damage [24].”

Line 204-205: I would remove the sentence “L*, a* and b* are the colour coordinates reflecting lightness, redness and yellowness”

Line 230: Please put “p” in italics

Line 247 (Table 1): Should capital letter (N) instead of small (n) be used to indicate the total number of animals?

Table 1: Could you use in the first row “Category” instead of “Numbers? Please put a space between Transports and (n)

Table 1: Please write “Meat quality” instead of “Meat Quality” 

Table 2: Please delete the second decimals also from Mix 0, ham - scores 3 and 4

Table 3 Can the caption be shortened to “…Temperature and pH1 were determined 45 min post-mortem, the other values after 48h post-mortem.” Maybe also the start “Within a column and variable,” could be removed, leaving the sentence into a form of “Ls Means and standard errors without a common superscript letter differ”   

I would replace “*p < 0.05, **p < 0.05, ***p < 0.001” with “*statistically significant difference (the name of test, p < 0.05)” 

Line 285: Please start the sentence with “Though/However, pigs which were…” and use only one decimal in the incidence of lesions

Line 292: I would prefer saying “social memory, i.e. the ability to recognise (recent) groupmates”

Line 298: Please replace “the regrouped pigs” with “the pigs regrouped”

Line 328: Could you say “Non-regrouping of pigs is the best option concerning animal welfare and meat quality.”?

Tables 2 and 3: Please check what the journal says about the explanations for superscript letters; I think they should be placed below the table, not in the heading.
